# Risk factors of papillary thyroid microcarcinoma that predispose patients to local recurrence

**Krzysztof Kaliszewski**[1]*, **Dorota Diakowska**[2], **Marta Rzeszutko**[3], **Łukasz Nowak**[4], **Michał Aporowicz**[5], **Beata Wojtczak**[1], **Krzysztof Sutkowski**[1], **Jerzy Rudnicki**[1]

**1** Department of General, Minimally Invasive and Endocrine Surgery, Wroclaw Medical University, Wroclaw, Poland, **2** Department of Nervous System Diseases, Faculty of Health Science, Wroclaw Medical University, Wroclaw, Poland, **3** Department of Pathomorphology, Wroclaw Medical University, Wroclaw, Poland, **4** Department of Urology and Urological Oncology, Wroclaw Medical University, Wroclaw, Poland, **5** Department of Surgery Didactics, Wroclaw Medical University, Wroclaw, Poland

\* krzysztofkali@wp.pl

## Abstract

### Background

Currently, less aggressive treatment or even active surveillance of papillary thyroid microcarcinoma (PTMC) is widely accepted and recommended as a therapeutic management option. However, there are some concerns about these approaches. We investigated whether there are any demographic, clinical and ultrasound characteristics of PTMC patients that are easy to obtain and clinically available before surgery to help clinicians make proper therapeutic decisions.

### Methods

We performed a retrospective chart review of 5,021 patients with thyroid tumors surgically treated in one center in 2008–2018. Finally, 182 (3.62%) PTMC patients were selected (158 (86.8%) females and 24 (13.2%) males, mean age 48.8±15.4 years). We analyzed the disease-free survival (DFS) time of the PTMC patients according to demographic and histopathological parameters. Univariate and multivariate logistic regression analyses were used to assess the relationships of demographic, clinical and ultrasound characteristics with aggressive histopathological features.

### Results

Age ≥55 years, hypoechogenicity, microcalcifications, irregular tumor shape, smooth margins and high vascularity significantly increased the risk for minimal extrathyroidal extension (minETE), lymph node metastasis (LNM), and capsular and vascular invasion (p<0.0001). Multivariate logistic regression analysis demonstrated a statistically significant risk of LNM (OR = 5.98, 95% CI: 2.32–15.38, p = 0.0002) and trends toward significantly higher rates of minETE and capsular and vascular invasion (OR = 2.24, 95% CI: 0.97–5.19, p = 0.056) in patients ≥55 years than in their younger counterparts. The DFS time was significantly

corresponding author and may be shared if
necessary.

**Funding:** This study was supported by Internal
Grant for Science Development of Wroclaw
Medical University in Poland, Grant Number: SUB.
B110.20.015.

**Competing interests:** The authors have declared
that no competing interests exist.

shorter in patients ≥55 years (p = 0.015), patients with minETE and capsular and vascular invasion (p = 0.001 for all), patients with tumor size >5 mm (p = 0.021), and patients with LNM (p = 0.002).

## Conclusions

The absence of microcalcifications, irregular tumor shape, blunt margins, hypoechogenicity and high vascularity in PTMC patients below 55 years and with tumor diameters below 5 mm may allow clinicians to select individuals with a low risk of local recurrence so that they can receive less aggressive management.

## Introduction

Currently, less aggressive surgical treatment or even active surveillance of papillary thyroid microcarcinoma (PTMC) is widely accepted and recommended as a therapeutic management option [1, 2]. These approaches are also popular among specialists because the American Thyroid Association (ATA) guidelines adopted and recommended them for selected PTMC patients [3]. However, some authors have raised concerns about these approaches [4], such as lymph node metastasis (LNM). Indeed, some years ago, LNM was treated as a poor prognostic indicator for only follicular thyroid cancer (FTC). Currently, the presence and accurately even number of cervical LNMs, also in young papillary thyroid cancer (PTC) patients (below 55 years), is recognized as a risk factor for poor prognosis [4]. The same situation concerns PTMC. Adam et al. [4] suggested that LNM predicts poor prognosis and decreased overall survival (OS) in all PTC patients, including PTMC patients. The next fundamental concern pertains to minimal extrathyroidal extension (minETE), even though it is detected only on histopathological examination and has been removed from the definition of pT3 disease and therefore has no impact on either pT category or overall stage. These two features of PTMC seem extremely important, especially when making therapeutic decisions.

From a clinical and pragmatic point of view, when making decisions regarding less aggressive surgery or active surveillance, it is difficult to assess the status of neck lymph nodes, the presence of minETE, and capsular and vascular invasion. In short, these characteristics, which are mandatory for proper decisions, cannot be ascertained without surgery [5]. Srivastava et al. [6] also remarked that it is difficult to directly assess the natural course of PTC.

In clinical practice, especially in surgical practice, it is unjustified to manage patients diagnosed with PTMC who have undiscovered capsular and vascular invasion, minETE or LNM by observation only and not radical treatment.

Therefore, we investigated whether there are any demographic, clinical and ultrasound characteristics of PTMC that are easy to obtain and clinically available prior to surgery to help clinicians make proper therapeutic decisions.

## Materials and methods

All procedures were in accordance with the ethical standards of the institutional and/or national research committee and with the 1964 Declaration of Helsinki and its later amendments or comparable ethical standards. We obtained verbal consent from the participants instead of written consent because the data were analyzed retrospectively and anonymously on the basis of medical records. The authors did not have access to identifying patient

information or direct access to the study participants. Our study protocol was approved by the Bioethics Committee of Wroclaw Medical University, Poland (Signature number: KB-783/2017).

We performed retrospective chart reviews of 5,021 patients with thyroid tumors who were admitted to and surgically treated at the Department of General, Gastrointestinal and Endocrine Surgery of Wroclaw Medical University (Poland) between 2008 and 2018. Among the patients, 596 (11.87%) had thyroid malignancies, of whom we selected 523 (10.41%) patients with PTC. Finally, 182 (3.62%) patients had tumors with a maximum diameter of 1.0 cm described on the basis of histopathological examination and were thereby classified as PTMC. According to the 8[th] Edition American Joint Committee on Cancer (AJCC)/tumor-node-metastasis (TNM) Classification [7], we excluded all tumors with a maximum size of 1.0 cm demonstrating gross ETE into the strap muscles, qualified by our pathologists not as T1a but as T3b tumors [7]. Finally, we included 158 (86.8%) females and 24 (13.2%) males, with a mean age of 48.8 $\pm$ 15.4 years, in the study group. All of the patients were staged in accordance with the TNM staging criteria proposed by the AJCC (8[th] Edition) [7]. Preoperative thyroid ultrasonography, ultrasound guided-fine-needle aspiration biopsy (UG-FNAB) and cytological examinations were performed in all cases. All suspected lesions were biopsied, and on this basis, presurgical multifocality and bilaterality were established. All cytological specimens were evaluated and assessed according to the Bethesda System for Reporting Thyroid Cytopathology (TBSRTC) criteria [8]. All patients included in the study underwent radical surgery (total thyroidectomy with central lymph node dissection). All individuals diagnosed with PTMC after surgery were routinely sent to the Oncology Center in Gliwice (Poland) for consultation and possible adjuvant radioiodine (RAI) therapy. The patients with pN0 features did not receive adjuvant RAI therapy, and the individuals with pN1 features received a single dose of ablated RAI in therapeutic amounts. All patients agreed to strict follow-up and remained under observation. The surgical tissue specimens were fixed in 10% buffered formalin and diagnosed histopathologically. Representative blocks were selected. Lesions with a maximum diameter of 1.0 cm or less without gross ETE were processed in their entirety; however, a minimum of 5–8 blocks were taken from each lesion. Serial sectioning and careful cutting of the representative tissue sample were performed, followed by routine methods of specimen processing. The sections were cut into 4 μm-thick sections, from which conventional hematoxylin and eosin (H&E) staining sections were prepared. The H&E sections were evaluated by two experienced thyroid lesion pathologists to confirm the diagnosis, features of the tumor and extent of the malignant process. The variables included patient age at diagnosis, sex, tumor size, LNM, distant metastasis at initial treatment, echogenicity, multifocality, bilaterality, number of foci, microcalcifications, tumor shape, margins, pathological vascularity, capsular end vascular invasion and minETE. The exclusion criteria included gross ETE, secondary tumors, previous head and neck surgery, head and neck radiation exposure, and incomplete clinical or histopathological data. Next, we analyzed demographic and ultrasound features and compared them with histopathological characteristics. Some of them are considered pathognomonic for aggressive PTMC entities [9]. Univariate and multivariate logistic regression analyses were used to determine the relationships of demographic and ultrasound features with histopathological characteristics such as minETE, capsular and vascular invasion and LNM. The Kaplan-Meier method and log-rank test were performed to compare the disease-free survival (DFS) time of patients divided into two groups according to demographic and selected histopathological parameters. For the purpose of this study, we assessed DFS time as the length of time (months) after primary surgery during which the patient survived without any structural evidence of disease. A p-value less than 0.05 was considered statistically significant. The follow-up

duration was a minimum of 12 months and maximum of 144 months, with a median interval of 70 months.

## Statistical analysis

Descriptive data are shown as the number of observations and percentages for qualitative variables or the means ± standard deviations (SDs) and 95% confidence intervals (95% CIs) for quantitative variables. The distribution of data for the parameter "Age" was analyzed using the Shapiro-Wilk normality test. Categorical variables were compared using Pearson chi-square, Fisher's exact and chi-square Wald tests, and continuous variables were compared by Student's t test. The odds ratios (ORs) for the relationship between histopathological parameters and demographic or ultrasound features were calculated using univariate and multivariate logistic regression analyses. $P \geq 0.1$ was the exclusion criterion for multivariate logistic regression analysis. Considering the small number of PTMC cases, p-values arising from logistic regression were confirmed by Fisher's exact test using 2-way contingency tables.

Two-tailed p-values $\leq 0.05$ were considered statistically significant. Statistical analysis was performed using STATISTICA v.13.3 (Tibco Inc., OK, USA), and figures were created using MS Excel 2016.

## Results

### Demographic and clinicopathological characteristics of the study group

As shown in Table 1, 132 PTMC patients were histopathologically diagnosed with stage I and 50 patients were diagnosed with stage II disease. All of the patients had stage pT1a tumors (Table 1). We calculated the occurrence of five predictors of poor PTMC prognosis, namely, minETE, capsular invasion, vascular invasion, LNM, and tumor size greater than 5 mm, for each patient in the study group. The majority of patients either did not exhibit the analyzed factors (n = 63, 34.6%), or all analyzed predictors were present (n = 51, 28.0%) (Table 1).

### Risk factors for minETE, capsular invasion and vascular invasion

In univariate logistic regression analysis, age 55 years or older, hypoechogenicity, microcalcification, irregular tumor shape, smooth margins and high vascularity significantly increased the risk of minETE and capsular and vascular invasion (Tables 2–4). The multivariate logistic regression analysis demonstrated significantly higher rates of capsular invasion (OR = 2.33, ±95%:1.02–5.34, p = 0.042) in patients aged 55 years or older (Table 3). A trend toward a significantly higher prevalence of minETE in older patients (OR = 2.24, +95% CI: 0.97–5.19, p = 0.056) was also observed (Table 2). In contrast, the presence of unfavorable ultrasound features did not affect the risk of negative histopathological results (Tables 2 and 3). No significant differences were found in the multivariate logistic regression for vascular invasion (Table 4).

### Risk factors for tumor size equal to or greater than 5 mm and presence of LNM

As shown in Table 3, age 55 years or older and all evaluated ultrasound features were significantly associated with tumors equal to or greater than 5 mm in diameter in PTMC patients. Then, a multivariate logistic regression model demonstrated that the probability of a tumor size greater than 5 mm significantly increased with high vascularity (OR = 39.32, 95% CI: 2.26–683.54, p = 0.011) (Table 5).

**Table 1. Demographic, clinical and histopathological characteristics of PTMC patients.**

| Variable | Total group (n = 182) |
|---|---|
| Sex: | |
| Female | 158 (86.8) |
| Male | 24 (13.2) |
| Age (years) | 48.8 ± 15.4 |
| Age: | |
| <55 years | 107 (58.8) |
| ≥55 years | 75 (41.2) |
| Presurgical diagnosis of multifocality: | |
| Yes | 22 (12.1) |
| No | 160 (87.9) |
| Presurgical diagnosis of bilaterality: | |
| Yes | 5 (2.8) |
| No | 177 (97.2) |
| Number of foci: | |
| 1 | 121 (66.5) |
| 2 | 40 (22.0) |
| 3 | 21 (11.5) |
| 4 | 0 (0.0) |
| minETE: | |
| Yes | 69 (37.9) |
| No | 113 (62.1) |
| pTNM: | |
| I | 132 (72.5) |
| II | 50 (27.5) |
| III | 0 (0.0) |
| IV | 0 (0.0) |
| pT1a | 182 (100) |
| pN: | |
| N0 | 109 (59.9) |
| N0a | 34 (18.7) |
| N0b | 75 (41.2) |
| N1 | 73 (40.1) |
| N1a | 73 (40.1) |
| N1b | 0 (0.0) |
| pM: | |
| M0 | 171 (94.0) |
| M1 | 0 (0.0) |
| Mx | 11 (6.0) |
| Number of predictors of a poor prognosis[#]: | |
| 0 | 63 (34.6) |
| 1 | 35 (19.2) |
| 2 | 14 (7.7) |
| 3 | 5 (2.7) |
| 4 | 14 (7.7) |
| 5 | 51 (28.0) |

Descriptive data are presented as the number of observations (percent) or the mean ± SD.

PTMC: papillary thyroid microcarcinoma; minETE: minimal extrathyroidal extension detected only on histological examination

[#]: The following predictors were analyzed: minETE, capsular invasion, vascular invasion, LNM, tumor size greater than 5 mm

**Table 2. Univariate and multivariate logistic regression analysis of demographic and ultrasound features as risk factors for minETE in PTMC patients.**

| | Extrathyroidal extension | | | | | | | |
|---|---|---|---|---|---|---|---|---|
| | No (n = 113) | Yes (n = 69) | Univariate analysis | | | Multivariate analysis | | |
| | | | OR | ± 95% CI | p-value | OR | ± 95% CI | p-value |
| Sex: | | | 1.77 | 0.74–4.20 | 0.258 | - | - | - |
| Female | 101 (89.4) | 57 (82.6) | | | | | | |
| Male | 12 (10.6) | 12 (17.4) | | | | | | |
| Age: | | | 3.07 | 1.64–5.73 | 0.0004* | 2.24 | 0.97–5.19 | 0.056# |
| < 55 years | 78 (69.0) | 29 (42.0) | | | | | | |
| ≥ 55 years | 35 (31.0) | 40 (58.0) | | | | | | |
| Hypoechogenicity: | | | 10.86 | 5.28–22.33 | <0.0001* | 5.11 | 0.49–53.20 | 0.169 |
| No | 83 (73.4) | 14 (20.3) | | | | | | |
| Yes | 30 (26.6) | 55 (79.7) | | | | | | |
| Microcalcification: | | | 10.70 | 4.68–24.47 | <0.0001* | 1.93 | 0.54–6.98 | 0.308 |
| No | 66 (58.4) | 8 (11.6) | | | | | | |
| Yes | 47 (41.6) | 61 (88.4) | | | | | | |
| Tumor shape: | | | 9.16 | 4.56–18.41 | <0.0001* | 1.46 | 0.19–11.08 | 0.711 |
| regular | 83 (73.4) | 16 (23.2) | | | | | | |
| irregular | 30 (26.6) | 53 (76.8) | | | | | | |
| Sharp margins: | | | 9.59 | 4.76–19.33 | <0.0001* | 1.46 | 0.18–11.09 | 0.712 |
| No | 84 (74.3) | 16 (23.2) | | | | | | |
| Yes | 29 (25.7) | 53 (76.8) | | | | | | |
| Vascularity: | | | 9.96 | 4.91–20.22 | <0.0001* | 0.96 | 0.10–9.03 | 0.975 |
| Low | 83 (73.4) | 15 (21.7) | | | | | | |
| High | 30 (26.6) | 54 (78.3) | | | | | | |

Descriptive data are presented as numbers (percentages), and the results were calculated by the chi-square Wald test.

*: statistically significant

#: tendency toward a statistically significant value

Univariate logistic regression analysis showed that age 55 years or older and all examined ultrasound features significantly increased the rate of LNM in PTMC patients (Table 4). Multivariate logistic regression analysis confirmed a significantly higher risk of LNM in patients aged 55 years or older than in their younger counterparts (OR = 5.98, 95% CI: 2.32–15.38, p = 0.0002) (Table 6).

Additionally, we performed an analysis of the presurgical diagnosis of PTMC for the period of 2008–2018. The frequency of PTMC patients diagnosed before surgery significantly increased during those 12 years of observation (p = 0.018) (Fig 1).

## Relationship between time to cancer recurrence and patient demographics or selected histopathological parameters

To analyze the associations between DFS time and patient demographics or histopathological variables, we selected PTMC patients according to sex, age at the time of diagnosis, minETE, capsular and vascular invasion, tumor size and LNM. As shown in Fig 2A, the DFS time was significantly shorter in patients with aged 55 years or older than in younger individuals (p = 0.015). Sex was a nonsignificant factor for cancer recurrence (p = 0.359). Kaplan-Meier curves revealed a significantly shorter DFS time for patients with minETE (p = 0.001) (Fig 2B); the same results were obtained for capsular and vascular invasion (p = 0.001). The DFS time

**Table 3. Univariate and multivariate logistic regression analysis of demographic and ultrasound features as risk factors for capsular invasion in PTMC patients.**

| | Capsular invasion | | | | | | | |
|---|---|---|---|---|---|---|---|---|
| | No (n = 113) | Yes (n = 69) | Univariate analysis | | | Multivariate analysis | | |
| | | | OR | ± 95% CI | p-value | OR | ± 95% CI | p-value |
| Sex: | | | 2.15 | 0.90–5.12 | 0.078 | - | - | - |
| Female | 102 (90.3) | 56 (81.2) | | | | | | |
| Male | 11 (9.7) | 13 (18.8) | | | | | | |
| Age: | | | 3.07 | 1.64–5.73 | 0.0003* | 2.33 | 1.02–5.34 | 0.042* |
| < 55 years | 78 (69.0) | 29 (42.0) | | | | | | |
| ≥ 55 years | 35 (31.0) | 40 (58.0) | | | | | | |
| Hypoechogenicity: | | | 9.52 | 4.70–19.28 | <0.0001* | 4.85 | 0.47–49.32 | 0.178 |
| No | 82 (72.6) | 15 (21.7) | | | | | | |
| Yes | 31 (27.4) | 54 (78.3) | | | | | | |
| Microcalcification: | | | 9.02 | 4.08–19.96 | <0.0001* | 1.65 | 0.47–5.83 | 0.432 |
| No | 65 (57.5) | 9 (13.0) | | | | | | |
| Yes | 48 (42.5) | 60 (87.0) | | | | | | |
| Tumor shape: | | | 8.09 | 4.07–16.06 | <0.0001* | 1.50 | 0.18–12.54 | 0.706 |
| regular | 82 (72.6) | 17 (24.6) | | | | | | |
| irregular | 31 (27.4) | 52 (75.4) | | | | | | |
| Sharp margins: | | | 7.49 | 3.80–14.76 | <0.0001* | 0.86 | 0.13–5.72 | 0.876 |
| No | 82 (72.6) | 18 (26.1) | | | | | | |
| Yes | 31 (27.4) | 51 (73.9) | | | | | | |
| Vascularity: | | | 8.86 | 4.37–17.56 | <0.0001* | 1.08 | 0.10–11.61 | 0.947 |
| Low | 82 (72.6) | 16 (23.2) | | | | | | |
| High | 31 (27.4) | 53 (76.8) | | | | | | |

Descriptive data are presented as numbers (percentages), and the results were calculated by the chi-square Wald test.

*: statistically significant

was also significantly shorter in patients with tumor size >5 mm and LNM (p = 0.021 and p = 0.002, respectively) (Fig 2C and 2D).

## Discussion

We assessed the demographic, clinical, ultrasound and histopathological characteristics of PTMC patients and showed that some of the clinical features that are available prior to surgery are associated with a higher risk of local recurrence and can help guide clinicians in making safe and proper therapeutic decisions. These estimated pre-surgically available ultrasound characteristics are associated with shorter disease-free survival times in PTMC patients and with histopathological features responsible for the potentially aggressive course of PTMC [9]. Thus, the results of our study might be useful for decisions regarding less aggressive surgical treatment (such as hemithyroidectomy) or even active surveillance, which are currently widely used. If we know that the risk of local recurrence is potentially low, then less aggressive management such as hemithyroidectomy may be justified and recommended. However, we must emphasize that we estimated only the risk factors for local recurrence and disease-free survival time, not OS time. We did not perform this analysis because the time of observation was too short to obtain reliable results. All of our patients, regardless of the analyzed parameters and disease-free survival time, are still alive. Additionally, we found that capsular and vascular invasion, minETE and LNM increased with patient age, which is not concordant with some

**Table 4. Univariate and multivariate logistic regression analysis of demographic and ultrasound features as risk factors for vascular invasion in PTMC patients.**

| | Vascular invasion | | | | | | | |
| | No (n = 114) | Yes (n = 68) | Univariate analysis | | | Multivariate analysis | | |
| | | | OR | ± 95% CI | p-value | OR | ± 95% CI | p-value |
|---|---|---|---|---|---|---|---|---|
| Sex: | | | 1.82 | 0.77–4.32 | 0.169 | - | - | - |
| Female | 102 (89.5) | 56 (82.4) | | | | | | |
| Male | 12 (10.5) | 12 (17.6) | | | | | | |
| Age: | | | 2.91 | 1.56–5.42 | 0.0006* | 1.86 | 0.82–4.25 | 0.133 |
| < 55 years | 78 (68.4) | 29 (42.7) | | | | | | |
| ≥ 55 years | 36 (31.6) | 39 (57.3) | | | | | | |
| Hypoechogenicity: | | | 9.05 | 4.47–18.30 | <0.0001* | 5.25 | 0.51–53.90 | 0.159 |
| No | 82 (71.9) | 15 (22.1) | | | | | | |
| Yes | 32 (28.1) | 53 (77.9) | | | | | | |
| Microcalcification: | | | 10.31 | 4.51–23.55 | <0.0001* | 2.66 | 0.74–9.51 | 0.128 |
| No | 66 (57.9) | 8 (11.8) | | | | | | |
| Yes | 48 (42.1) | 60 (88.2) | | | | | | |
| Tumor shape: | | | 7.69 | 3.87–15.23 | <0.0001* | 1.35 | 0.16–11.44 | 0.779 |
| regular | 82 (71.9) | 17 (25.0) | | | | | | |
| irregular | 32 (28.1) | 51 (75.0) | | | | | | |
| Sharp margins: | | | 7.11 | 3.62–13.99 | <0.0001* | 0.79 | 0.12–5.13 | 0.800 |
| No | 82 (71.9) | 18 (26.5) | | | | | | |
| Yes | 32 (28.1) | 50 (73.5) | | | | | | |
| Vascularity: | | | 8.32 | 4.16–16.66 | <0.0001* | 0.86 | 0.08–9.24 | 0.901 |
| Low | 82 (71.9) | 16 (23.5) | | | | | | |
| High | 32 (28.1) | 52 (76.5) | | | | | | |

Descriptive data are presented as numbers (percentages), and the results were calculated by the chi-square Wald test.

*: statistically significant

other analyses in several aspects [5]. We revealed that older PTMC patients had an increased predisposition for adverse histopathological features. Although some studies have tried to determine the clinicopathological status of PTC patients prior to surgery, we believe our results will improve the accuracy of presurgical decisions. Borsò et al. [10] introduced a radio-guided occult localization technique to identify the cervical recurrence of well-differentiated thyroid cancers (WDTCs), but in the opinion of some other authors, this technique is not sufficient for identifying the presence of LNM and determining recurrence before surgery [5]. The same authors stated that the rate of LNM decreases with age in patients with PTC [5], suggesting that LNM may disappear spontaneously in some middle-aged PTC patients due to delayed treatment. They explain that younger patients have more effective immunological reactions, which may lead to lesion shrinkage and the spontaneous resolution of metastatic lymph nodes. The authors concluded that the activation of autoimmunological reactions leads to a lower risk and a better survival of patients suffering from PTC. Therefore, if LNMs may disappear spontaneously, then we may treat this phenomenon as supporting evidence that some PTMCs present an indolent clinical course and are not "true" cancers [9]. Moreover, if the LNMs of some PTCs disappear, then we can treat this phenomenon as another argument to support the less aggressive treatment of this malignancy. Lamartina et al. [11] performed a retrospective study and revealed that in 3.8% of PTMC patients, clinically suspicious LNMs disappeared during the follow-up period. These results are interesting and, as noted above, could allow clinicians to consider less aggressive management for some PTMC patients.

**Table 5. Univariate and multivariate logistic regression analyses of demographic and ultrasound features as risk factors for tumors equal to or greater than 5 mm in diameter in PTMC patients.**

| | | | Tumor size | | | | | |
|---|---|---|---|---|---|---|---|---|
| | <5 mm (n = 79) | ≥5 mm (n = 103) | Univariate analysis | | | Multivariate analysis | | |
| | | | OR | 95% CI | p-value | OR | 95% CI | p-value |
| Sex: | | | 2.03 | 0.79–5.18 | 0.131 | - | - | - |
| Female | 72 (91.1) | 86 (83.5) | | | | | | |
| Male | 7 (8.9) | 17 (16.5) | | | | | | |
| Age: | | | 2.24 | 1.21–4.16 | 0.009* | 1.64 | 0.47–5.76 | 0.434 |
| < 55 years | 55 (69.6)) | 52 (50.5) | | | | | | |
| ≥ 55 years | 24 (30.4) | 51 (49.5) | | | | | | |
| Hypoechogenicity: | | | 69.03 | 22.73–209.61 | <0.0001* | 11.86 | 0.63–222.41 | 0.095 |
| No | 75 (94.9) | 22 (21.4) | | | | | | |
| Yes | 4 (5.1) | 81 (78.6) | | | | | | |
| Microcalcification: | | | 15.01 | | <0.0001* | 1.01 | 0.24–4.18 | 0.988 |
| No | 58 (73.4) | 16 (15.5) | | 7.23–31.17 | | | | |
| Yes | 21 (26.6) | 87 (84.5) | | | | | | |
| Tumor shape: | | | 46.17 | | <0.0001* | 4.56 | 0.11–176.41 | 0.412 |
| Regular | 74 (93.7) | 25 (24.3) | | 16.79–126.96 | | | | |
| Irregular | 5 (6.3) | 78 (75.7) | | | | | | |
| Sharp margins: | | | 43.83 | | <0.0001* | 4.87 | 0.10–178.10 | 0.417 |
| No | 5 (6.3) | 77 (74.8) | | | | | | |
| Yes | 74 (93.7) | 26 (25.2) | | 15.98–120.21 | | | | |
| Vascularity: | | | 93.27 | | <0.0001* | 39.32 | 2.26–683.54 | 0.011* |
| Low | 76 (96.2) | 22 (21.4) | | | | | | |
| High | 3 (3.8) | 81 (78.6) | | 26.82–324.32 | | | | |

Descriptive data are presented as numbers (percentages), and the results were calculated by the chi-square Wald test.

PTMC: papillary thyroid microcarcinoma

*: statistically significant

Ito et al. [12] revealed that, in some follow-up periods, 4.2–16.6% of PTMCs decreased in diameter compared to that at the beginning of the observation. Miyauchi et al. [13] performed an interesting study in which they described some differences in tumor size in PTMC, noting that the size decreased in accordance with the age of the observed PTMC patients. Generally, they noted that an average of 17% of PTMC patients had decreasing tumor sizes during the observation time, with 15, 21 and 13% of the patients experiencing tumor shrinkage, respectively, according to age distribution (i.e., ≤40, 41–60, and >60 years). In another study, among 300 patients with PTMC, it was estimated that 3% had decreased tumor dimensions during the follow-up period [14]. These observations of PTMC diameter are connected with potential tumor regression; thus, these findings might also support less aggressive treatment options. We found the results of the second group of observed PTMC patients to be particularly interesting. Nevertheless, despite the 3.8% of PTMC patients with disappeared LNMs reported by Lamartina et al. [11] or the 3% of PTMC patients in which decreasing tumor dimensions were observed [14], we cannot choose less aggressive treatment or active surveillance options for all PTMC patients. As such, we believe that the results of our analysis might be useful in the clinical management of all PTMC patients.

In contrast with our results, some authors have reported that younger patients have a higher rate of LNM than older and middle-aged patients [15]. In such situations, other dilemmas

**Table 6. Univariate and multivariate logistic regression analyses of demographic and ultrasound features as risk factors for LNM in PTMC patients.**

| | pN0 (n = 109) | pN1 (n = 73) | OR | 95% CI | p-value | OR | 95% CI | p-value |
|---|---|---|---|---|---|---|---|---|
| | | | | Univariate analysis | | | Multivariate analysis | |
| Sex: | | | 2.34 | 0.98–5.62 | 0.072 | - | - | - |
| Female | 99 (90.8) | 59 (80.8) | | | | | | |
| Male | 10 (9.2) | 14 (19.2) | | | | | | |
| Age: | | | 5.23 | 2.74–9.95 | <0.0001* | 5.98 | 2.32–15.38 | 0.0002* |
| < 55 years | 81 (74.3) | 26 (35.6) | | | | | | |
| ≥ 55 years | 28 (25.7) | 47 (64.4) | | | | | | |
| Hypoechogenicity: | | | 13.45 | 6.48–27.92 | <0.0001* | 4.07 | 0.37–44.37 | 0.247 |
| No | 83 (76.2) | 14 (19.2) | | | | | | |
| Yes | 26 (23.8) | 59 (80.8) | | | | | | |
| Microcalcification: | | | 15.04 | 6.31–35.87 | <0.0001* | 1.46 | 0.40–5.37 | 0.558 |
| No | 67 (61.5) | 7 (9.6) | | | | | | |
| Yes | 42 (38.5) | 66 (90.4) | | | | | | |
| Tumor shape: | | | 11.37 | 5.60–23.09 | <0.0001* | 1.17 | 0.16–8.68 | 0.872 |
| Regular | 83 (76.2) | 16 (21.9) | | | | | | |
| Irregular | 26 (23.8) | 57 (78.1) | | | | | | |
| Sharp margins: | | | 11.97 | 5.87–24.39 | <0.0001* | 1.19 | 0.20–9.21 | 0.834 |
| No | 25 (22.9) | 57 (78.1) | | | | | | |
| Yes | 84 (77.1) | 16 (21.9) | | | | | | |
| Vascularity: | | | 12.34 | 6.02–25.32 | <0.0001* | 1.36 | 0.15–12.49 | 0.782 |
| Low | 83 (76.2) | 15 (20.6) | | | | | | |
| High | 26 (23.8) | 58 (79.4) | | | | | | |

Descriptive data are presented as numbers (percentages), and the results were calculated by the chi-square Wald test.

LNM: lymph node metastasis; PTMC: papillary thyroid microcarcinoma

*: statistically significant

occur; younger patients with PTMC may choose less aggressive management due to, for example, cosmetic reasons or the necessity of postsurgical supplementation [16]. Therefore, it is clear that they should be evaluated very carefully before therapeutic decisions are made.

One of the Surveillance, Epidemiology, and End Results (SEER) studies, in which the authors analyzed the association between PTC patients' age and LNM, demonstrated that younger individuals had an increased LNM risk regardless of tumor size (T feature according to TNM staging) [17]. As Mehra et al. [18] suggested, the SEER database is useful for clinical analyses because it contains detailed tumor clinicopathological information, demographic data, patient surveillance parameters, and surgical and postsurgical treatment observations. On the other hand, some authors have asserted that the long observation time of PTMC patients may lead to progression and the ability of these small cancers to obtain malignant clinical features [19].

According to some analyses, even 60% of PTMC patients have LNM in the locoregional area at the time of diagnosis [20]. Thus, in accordance with the results of these studies, we may state that the majority of PTMCs might not be indolent precancers but true cancers [9, 19]. Therefore, in contrast to indolent and nonaggressive PTMC cases, some authors have described these small, theoretically silent malignancies as very aggressive tumors. Shawky et al. [21] presented a patient with seizures and dysphasia due to a cystic brain mass, which, after excision, revealed multifocal PTMC on histopathology, with the largest focus being 3 mm.

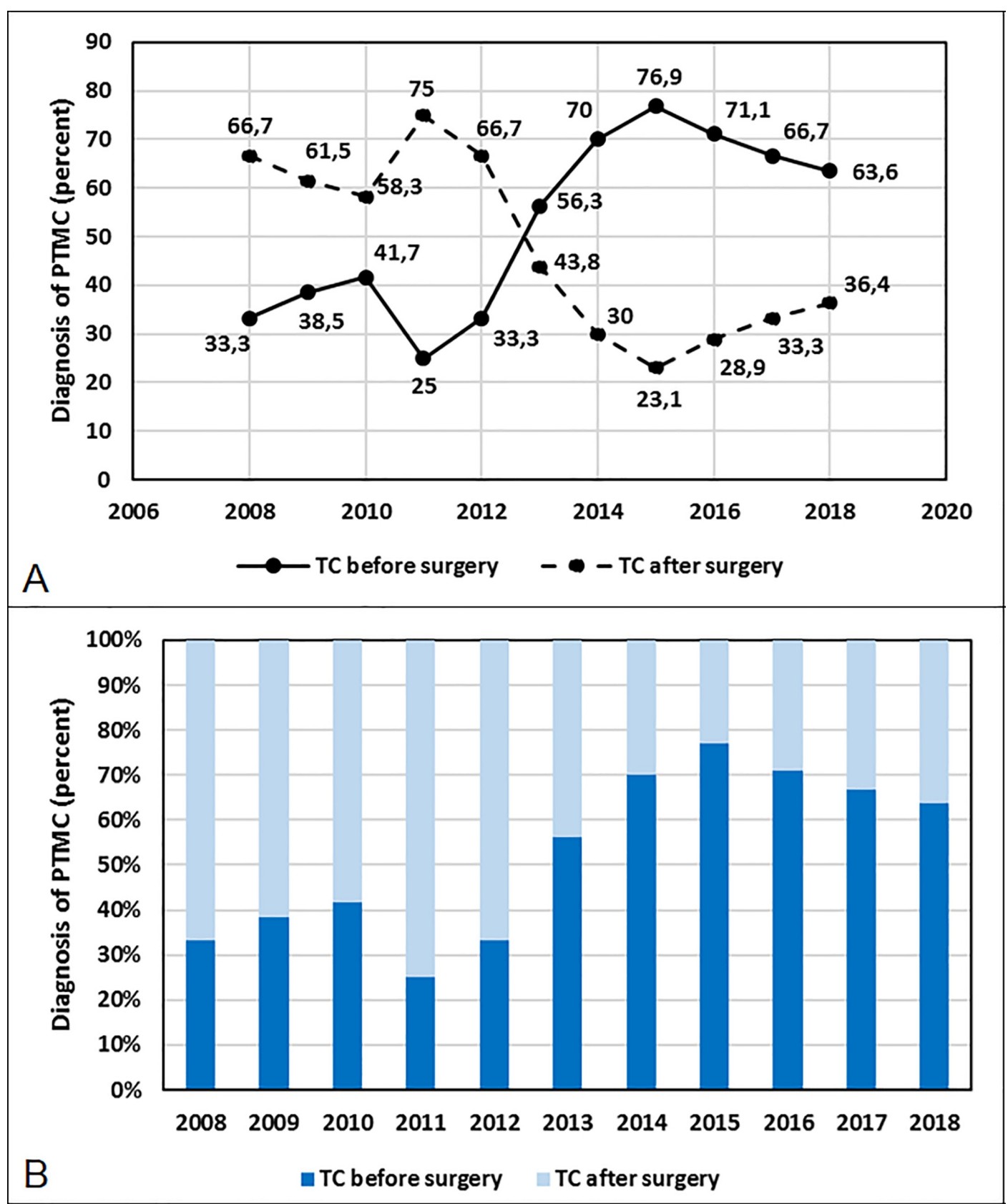

**Fig 1.** Percent of papillary thyroid microcarcinoma (PTMC) patients with a diagnosis of thyroid cancer (TC) before surgery and after surgery during the period of 2008–2018 (A, B). The frequency of the diagnosis of TC before surgery significantly increased during this period (p = 0.018).

These authors encouraged clinicians to select radical treatment and careful follow-up for such cases due to their high mortality. Yamashita et al. [22] reported a patient with occult PTC that was not detected before surgery or even in postsurgical histopathology. After resection of the upper neck mass, the presence of LNM of PTC was established. Subsequently, hyalinized images of the thyroid gland indicated that a small PTMC might have spontaneously disappeared [22]. Thus, because of this situation, we cannot treat all PTMCs with decreasing tumor size or disappearing LNM as absolutely indolent entities. Moreover, some authors have stated that the prevalence of occult thyroid cancers, which are small carcinomas in the thyroid diagnosed on secondary examination after the primary diagnosis of LNM or distant metastasis, makes these tumors more dangerous [23]. When a patient presents swelling in the neck lymph nodes, the probability of occult thyroid carcinoma is high [24]. The majority of occult thyroid

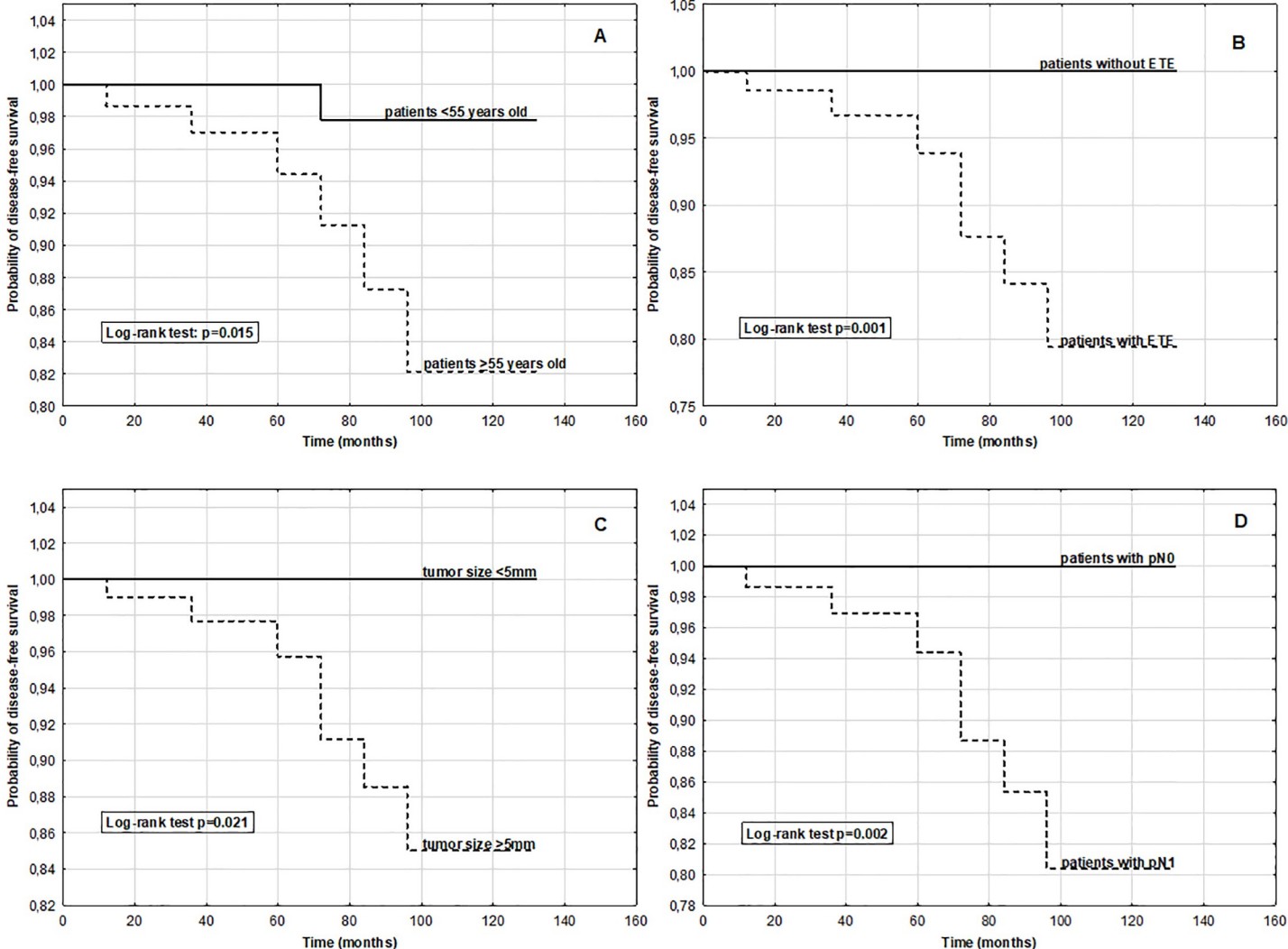

**Fig 2. Time of cancer recurrence for PTMC patients according to age and selected histopathological parameters.** Kaplan-Meier curves showed significantly lower disease-free survival times for patients aged ≥55 years (A), minimal extrathyroidal extension status (B), tumor size >5 mm (C) and lymph node metastasis (D).

cancers are equal to or smaller than 1 cm in diameter, and in the opinion of some authors, if LNM or distant metastasis is observed, their prognosis is poor [25].

While the World Health Organization (WHO) defined PTMC as a PTC with a diameter equal to or below 1.0 cm [26], some other researchers have divided PTMCs into two separate groups [27]. The first group contains PTMCs with LNM and distant metastasis or any other malignant tumor symptoms and is called the symptomatic group. The group without any cancer symptoms, such as LNM or distant metastasis, is called the asymptomatic group. According to the results of our previous [9] and current analysis, we suspect that PTMC patients in the first group should not be considered for less aggressive management. In our opinion, the main aspect of this approach is the proper division and qualification of PTMC patients to the appropriate group. In our study, we considered that the symptomatic group, which should not be observed but treated radically, should include PTMC patients with ultrasound features such as irregular tumor shape, smooth margins, high vascularity, hypoechogenicity and microcalcifications, especially older patients (≥55 years) with tumors equal to or greater than 5 mm in dimension. Others have suggested that active surveillance without surgery should be recommended for asymptomatic PTMC patients owing to their highly favorable prognosis [28]. Moreover, researchers have highlighted the poor prognosis of patients in the symptomatic group. For example, Sugitani et al. [14] presented patients with symptomatic PTMC and revealed that many of them had recurrent laryngeal nerve paralysis caused by the primary tumor, LNM or distant metastasis. Additionally, they estimated that 38% of PTMC patients had recurrence and that 8% died of primary disease. Almost 78% had poor prognostic factors such as ETE, extranodal invasion, LNM size above 20 mm and anaplastic components in the primary tumor. The 10-year disease-specific mortality rate of patients with these poor prognostic factors was 74%. Interestingly, the authors did not observe any death due to primary disease in the patients without these poor prognostic factors [14]. These results slightly support our hypothesis; however, as mentioned above, we did not perform an analysis of OS time because our observation period was not very long. Other authors have reported similar results [24], confirming that LNM should be treated as a poor prognostic factor. According to thyroid association guidelines [29], pathologically enlarged lymph nodes should be treated as a high-risk factor for PTC. However, as we speculated previously, clinically evident LNM is not a poor prognostic factor for PTMC. Theoretically, if LNM manifests before decisions regarding the further management of PTMC are made, then the situation is relatively simplified. When we performed our analysis, we especially focused on individuals in whom LNM was not clinically present prior to surgery. Some of these patients may be classified as having a good prognosis. Thus, we suggest that cases without clinically evident LNM but with an irregular shape, hypoechogenicity, high vascularity, smooth margins and microcalcifications should also be treated as high-risk PTMCs or symptomatic PTMCs. When the tumor is classified as high-risk PTMC or poor prognostic risk factors are evident, radical surgery should be recommended, not less aggressive management. After surgery, to prevent recurrence or metastasis, strict observation or further adjuvant therapy should be considered [22]. In such clinical situations, Yamashita et al. [22] recommended radioiodine ablation (RAI) in therapeutic doses. They performed 30 mCi 131-I ablation after surgery in all PTMC patients and did not note any recurrence or metastasis for 24 months after treatment. Generally, in our opinion, careful follow-up should be performed for all symptomatic PTMC patients after treatment due to the possibility of recurrence and metastasis.

Our study has some limitations that should be noted. First, this is a retrospective analysis, and access to some necessary details was limited, which resulted in the exclusion of several cases. Second, the analyzed data came from a single medical center; a multicenter analysis would be more reliable. Third, we performed our study and made conclusions after an analysis

of clinical, ultrasound and histopathological features, which only indirectly indicates PTMC cases with potentially poor or excellent prognoses. It is difficult to confirm our observations unless some patients who do not undergo surgery are followed until they die of other causes. Finally, the observation time of our study was not sufficient. Therefore, we did not perform an analysis of OS time. All analyzed patients are still alive and remain under observation.

## Conclusions

Considering the increasing prevalence of PTMC and the increasing selection of less aggressive treatment for this malignancy, it seems valuable to introduce accurate diagnostic criteria to aid in making proper therapeutic decisions. We suggest that the absence of microcalcifications, irregular tumor shape, blunt margins, hypoechogenicity and high vascularity in PTMC patients younger than 55 years with tumors less than 5 mm in diameter are associated with a low risk of local recurrence and lead to a better prognosis, which may allow clinicians to select less aggressive management strategies.

## Supporting information

**S1 Data.**
(XLS)

## Acknowledgments

The authors are grateful to all the staff at the study center who contributed to this work.

## Author Contributions

**Conceptualization:** Krzysztof Kaliszewski.

**Data curation:** Krzysztof Kaliszewski, Marta Rzeszutko, Łukasz Nowak, Michał Aporowicz, Beata Wojtczak.

**Formal analysis:** Krzysztof Kaliszewski, Dorota Diakowska.

**Funding acquisition:** Krzysztof Kaliszewski.

**Investigation:** Krzysztof Kaliszewski, Marta Rzeszutko, Łukasz Nowak, Michał Aporowicz, Beata Wojtczak.

**Methodology:** Krzysztof Kaliszewski, Dorota Diakowska.

**Project administration:** Krzysztof Kaliszewski.

**Resources:** Krzysztof Kaliszewski, Marta Rzeszutko, Łukasz Nowak, Beata Wojtczak, Krzysztof Sutkowski.

**Software:** Krzysztof Kaliszewski.

**Supervision:** Krzysztof Kaliszewski, Jerzy Rudnicki.

**Validation:** Krzysztof Kaliszewski, Dorota Diakowska, Łukasz Nowak, Michał Aporowicz.

**Visualization:** Krzysztof Kaliszewski, Marta Rzeszutko, Krzysztof Sutkowski, Jerzy Rudnicki.

**Writing – original draft:** Krzysztof Kaliszewski, Dorota Diakowska.

**Writing – review & editing:** Krzysztof Kaliszewski.

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
