## [Decision Letter · Decision Letter 0]

10 Nov 2020

PONE-D-20-23025

Assessment of the Ultrasound and Histopathological Features of Papillary Thyroid Microcarcinoma for Less Aggressive Management.

PLOS ONE

Dear Dr. Kaliszewski,

Thank you for submitting your manuscript to PLOS ONE. After careful consideration, we feel that it has merit but does not fully meet PLOS ONE’s publication criteria as it currently stands. Therefore, we invite you to submit a revised version of the manuscript that addresses the points raised during the review process.

Please accept the apologies for the long time the review process as taken.

Both reviewers consider that your Ms needs improvements both in the analysis as well as in the presentation and language.

Please address ALL the reviewers concerns in a point by point letter.

Take special attention to table 2 where risk factors for minETE, capsular invasion and vascular invasion should be individually analysed and presented. Also data showing how many patients had 1, 2, 3 or 4 risk factors are mandatory. In addition all the remaining points raised by the reviewers must be addressed. 

We look forward to receiving your revised manuscript.

Kind regards,

Paula Soares

Academic Editor

PLOS ONE

Journal Requirements:

"This study was supported by Internal Grant for Science Development of Wroclaw

Medical University in Poland, Grant Number: SUB.B110.20.015.".

i) We note that you have provided funding information that is not currently declared in your Funding Statement. However, funding information should not appear in the Acknowledgments section or other areas of your manuscript. We will only publish funding information present in the Funding Statement section of the online submission form.

ii) Please remove any funding-related text from the manuscript and let us know how you would like to update your Funding Statement. Currently, your Funding Statement reads as follows:

"The authors received no specific funding for this work.".

iii) Please include your amended statements within your cover letter; we will change the online submission form on your behalf.

Reviewers' comments:

Reviewer's Responses to Questions

**Comments to the Author**

1. Is the manuscript technically sound, and do the data support the conclusions?

Reviewer #1: Partly

Reviewer #2: Partly

2. Has the statistical analysis been performed appropriately and rigorously? 

Reviewer #1: I Don't Know

Reviewer #2: Yes

3. Have the authors made all data underlying the findings in their manuscript fully available?

Reviewer #1: Yes

Reviewer #2: No

4. Is the manuscript presented in an intelligible fashion and written in standard English?

Reviewer #1: No

Reviewer #2: No

5. Review Comments to the Author

Reviewer #1: Table 1 :

a) Presurgical diagnosis of multifocality and bilaterality, number of foci - how it was evaluated ? US ? FNA?

b) pTNM stage II - 4; pN1 -73 (pT1N1 = stage II)

c) cN ?

Table 2: risk factors for minETE, capsular invasion and vascular invasion should be individually analysed

To analyse disease free survival time it is necessary to know:

a) surgical treatment, CLN dissection

b) ablation/adjuvant radiodine therapy

c) recurrencies

d) the follow up time (min, max, medium)

none of these are mentioned and should be specified

The english must be improved

Reviewer #2: This paper contains some valuable data, but has a number of deficiencies that lessen the impact.

(1) The title is misleading. It should be something like: Papillary Thyroid microcarcinomas: risk factors for local recurrence

(2) The authors freely express their management opinion but do not discuss the rationale of the opposing point of view.. The topic is mildly controversial. It is unclear that more extensive surgery/treatment results in improved overall survival or that any treatment of PMTC improves overall survival. In fact, they do not offer any survival data, likely because there is no difference. Those promulgating the conservative approach to PMTC provide treatment when the lesions enlarge or LN metastases appear. So the argument, given their data, is whether local recurrence in PMTC is associated with decreased overall survival. They have presented no data on this point. Instead, they argue that local recurrence is a proxy for overall survival in PTC patients in general.

(3) The Kaplan Meier curves are so similar that one suspects that many of the patients that had a higher local recurrence rate had all 4 risk factors. Data showing how many patients had 1, 2, 3 or 4 risk factors needs to be presented.

(4). The discussion is rambling and very difficult to read. The formula that gives the best readability is: one paragraph of discussion that refers to a single paragraph of results, followed by one paragraph of conclusion. Each paragraph should begin with a sentence telling what the paragraph is about. Each paragraph should end with a conclusion, "Therefore..."

(5) Despite the certificate stating otherwise, this needs editing by a native English speaker with some scientific background.

6. PLOS authors have the option to publish the peer review history of their article (what does this mean?). If published, this will include your full peer review and any attached files.

Reviewer #1: No

Reviewer #2: No

---

## [Author Response · Author response to Decision Letter 0]

1 Dec 2020

Dear Editor and Reviewers,

First, we truly appreciate the opportunity to resubmit our revised manuscript. Thank you very much for considering our manuscript for potential publication in Plos One.

We would also like to thank you for the very thorough reviews, professional advice and constructive criticism, which were highly valuable for improving our paper. All suggestions regarding changes and improvements were very helpful to us, and we revised the manuscript according to the recommendations provided in the reviews. All changed and deleted portions of the manuscript are indicated with track changes. According to the reviewers’ instructions, we have corrected our manuscript point-by-point as follows.

Regarding the responses to the concrete comments by the reviewers, we corrected/changed the manuscript in the following ways:

Reviewer 1:

#Table 1:

a) “Presurgical diagnosis of multifocality and bilaterality, number of foci...”. Thank you very much for this remark. Clearly, the presurgical diagnosis of PTMC before surgery is sometimes difficult but very important. To establish the most accurate diagnosis of thyroid tumors before surgery in all patients, we performed ultrasonography and ultrasound-guided fine needle aspiration biopsy (UG-FNAB) of all suspected tumors, as indicated for this tumor type. If the results indicate two or more suspected malignant tumors, we performed UG-FNAB in all lesions. Occasionally, we make a cancer diagnosis in more than one thyroid tumor. Thus, in such cases, we know before surgery that the patient has multifocal bilateral PTMC. We added the following adequate explanation to the manuscript: “Preoperative thyroid ultrasonography, ultrasound guided-fine-needle aspiration biopsy (UG-FNAB) and cytological examinations were performed in all cases. All suspected lesions were biopsied, and on this basis, presurgical multifocality and bilaterality were established.”

b) We are immensely grateful for this remark. Indeed, we made a mistake when we assessed the patients according to TNM staging. According to the 8th Edition of the TNM Classification, in cases of well-differentiated papillary thyroid cancers, patients with N1 aged 55 years or above at the time of diagnosis are assigned to stage II. Thus, we reassessed and corrected the results in the table. Thank you.

c) We decided to not present the clinical classification of lymphatic nodules cN (pathologically changed lymphatic nodules) because these data were not strict and did not influence the results of our analysis. However, if you feel it is necessary to include these data, we will gladly reassess the medical records of all patients and include this clinical observation.

#Table 2:

We performed separate statistical analyses of minETE, capsular invasion and vascular invasion (Tables 2, 3, and 4).

In the Materials and methods, we added information regarding the following:

a) Surgical treatment, CLN dissection:

“All patients included in the study underwent radical surgery (total thyroidectomy with central lymph node dissection).”

b) Ablation/radioiodine therapy: The study was performed in the Surgical Department. After radical surgery, we sent all patients for consultation and potential qualification for RAI therapy at an oncology center in a different place.

“All individuals diagnosed with PTMC after surgery were routinely sent to the Oncology Center in Gliwice (Poland) for consultation and possible adjuvant radioiodine (RAI) therapy. The patients with pN0 features did not receive adjuvant RAI therapy, and the individuals with pN1 features received a single dose of ablated RAI in therapeutic amounts. All patients agreed to strict follow-up and remained under observation.”

c) Recurrence:

“For the purpose of this study, we assessed DFS time as the length of time (months) after primary surgery during which the patient survived without any structural evidence of disease. A p-value less than 0.05 was considered statistically significant.”

d) Follow-up time (min, max, medium):

“The follow-up duration was a minimum of 12 months and maximum of 144 months, with a median interval of 70 months.”

Although our manuscript was reviewed by American Journal Experts as recommended by the journal (we attached the certificate), after making our point-by-point corrections based on the suggestions by the reviewers, we sent our revised manuscript once again to AJE for proofreading (we have attached the new certificate). Thank you.

Reviewer 2:

First, we thank you very much for the opinion that our manuscript contains some valuable data. This is a very important acknowledgement for us. Thank you. Regarding your remarks, we corrected the text as follows:

1) We agree that the title is misleading. We changed the title to “Risk Factors of Papillary Thyroid Microcarcinoma that Predispose Patients to Local Recurrence.”

2) We absolutely agree that the topic undertaken in our study is controversial and that we do not have data confirming that a more aggressive treatment of PTMC provides better survival for all patients. Indeed, we analyzed only the risk factors for local recurrence and not overall survival time. In our analysis, all patients free of local recurrence were still alive, but the time of observation may be too short. Thus, we changed our discussion to avoid misleading readers. We drew hasty conclusions that local recurrence shortens the overall survival time, and as we know, in PTMC, this finding is not clearly established. Thank you for this valuable remark.

3) We included the data showing how many patients had 1, 2, 3 or 4 risk factors in Table 1 and the supportive data set.

4) We agree that the discussion was not well written; thus, we have rewritten and re-edited the discussion according to your suggestions. Thank you.

5) Although our manuscript was reviewed by American Journal Experts as recommended by the journal (we attached the certificate), after making our point-by-point corrections based on the suggestions by the reviewers, we sent our revised manuscript once again to AJE for proofreading (we have attached the new certificate). Thank you.

Best regards,

Krzysztof Kaliszewski MD, PhD

---

## [Decision Letter · Decision Letter 1]

21 Dec 2020

Risk Factors of Papillary Thyroid Microcarcinoma that Predispose Patients to Local Recurrence.

PONE-D-20-23025R1

Dear Dr. Kaliszewski,

We’re pleased to inform you that your manuscript has been judged scientifically suitable for publication and will be formally accepted for publication once it meets all outstanding technical requirements.

Kind regards,

Paula Soares

Academic Editor

PLOS ONE

Additional Editor Comments (optional):

Reviewers' comments:

Reviewer's Responses to Questions

**Comments to the Author**

1. If the authors have adequately addressed your comments raised in a previous round of review and you feel that this manuscript is now acceptable for publication, you may indicate that here to bypass the “Comments to the Author” section, enter your conflict of interest statement in the “Confidential to Editor” section, and submit your "Accept" recommendation.

Reviewer #2: All comments have been addressed

2. Is the manuscript technically sound, and do the data support the conclusions?

Reviewer #2: Yes

3. Has the statistical analysis been performed appropriately and rigorously? 

Reviewer #2: I Don't Know

4. Have the authors made all data underlying the findings in their manuscript fully available?

Reviewer #2: Yes

5. Is the manuscript presented in an intelligible fashion and written in standard English?

Reviewer #2: Yes

6. Review Comments to the Author

Reviewer #2: (No Response)

7. PLOS authors have the option to publish the peer review history of their article (what does this mean?). If published, this will include your full peer review and any attached files.

Reviewer #2: No

---

## [Editor Report · Acceptance letter]

23 Dec 2020

PONE-D-20-23025R1 

Risk Factors of Papillary Thyroid Microcarcinoma that Predispose Patients to Local Recurrence. 

Dear Dr. Kaliszewski:

I'm pleased to inform you that your manuscript has been deemed suitable for publication in PLOS ONE. Congratulations! Your manuscript is now with our production department. 

Kind regards, 

on behalf of

Dr. Paula Soares 

Academic Editor

PLOS ONE